# The Critical Balance Between Quiescence and Reactivation of Neural Stem Cells

**DOI:** 10.3390/biom15050672

**Published:** 2025-05-06

**Authors:** Adam M. Elkin, Sarah Robbins, Claudia S. Barros, Torsten Bossing

**Affiliations:** Peninsula Medical School, Faculty of Health, University of Plymouth, Plymouth PL6 8BU, UK; adam.elkin@plymouth.ac.uk (A.M.E.); sarah.robbins@plymouth.ac.uk (S.R.)

**Keywords:** neural stem cells, quiescence, reactivation, damage repair

## Abstract

Neural stem cells (NSC) are multipotent, self-renewing cells that give rise to all neural cell types within the central nervous system. During adulthood, most NSCs exist in a quiescent state which can be reactivated in response to metabolic and signalling changes, allowing for long-term continuous neurogenesis and response to injury. Ensuring a critical balance between quiescence and reactivation is required to maintain the limited NSC reservoir and neural replenishment throughout lifetime. The precise mechanisms and signalling pathways behind this balance are at the focus of current research. In this review, we highlight and discuss recent studies using *Drosophila*, mammalian and zebrafish models contributing to the understanding of molecular mechanisms underlying quiescence and reactivation of NSCs.

## 1. Introduction

The central nervous system (CNS) is a highly complex network allowing for many functions, including movement, learning, and memory. The billions of neurons and glia required to form this intricate system are derived from a relatively small population of progenitor cells termed neural stem cells (NSCs). Tissue stem cells have a remarkable self-renewal and multilineage capacity, allowing them to asymmetrically divide to generate diverse cell types during development, tissue homeostasis in adulthood, and during regenerative or damage repair processes [1,2,3]. This capacity is spatially and temporally regulated, allowing for the differing progeny that contribute to drive complex behaviours [4]. However, most NSCs during adulthood are maintained in a quiescent state, allowing for the precious, limited pool of NSCs to be preserved and reactivated as needed including following disease or injury [4,5,6]. Organisms with a high regenerative capacity such as zebrafish (*Danio rerio*) and salamanders utilise quiescent NSCs (qNSCs) to replenish neurons and glia within newly regenerated limbs [3]. The balance between quiescence and reactivation is critical, too little reactivation leads to impaired repair and growth, while excess reactivation could lead to uncontrolled growth and exhaustion of the NSC pool [3,5,7,8,9]. The mechanisms behind stem cell quiescence and reactivation may also be utilised by cancer cells, allowing quiescent cells to resist treatment targeted at rapidly dividing cells and re-activate the cell cycle at a later time-point [1,4]. While much progress has been achieved in the last years, knowledge of the complex mechanisms and signalling pathways controlling quiescence and reactivation of NSCs is far from comprehensive, therefore their better understanding is critical for the development of NSC-based therapies directed at tissue regeneration, repair and restoring function.

Different model systems have been utilised to study quiescence and reactivation of NSCs. A powerful system for investigating NSCs is the fruit fly *Drosophila* model. *Drosophila* NSCs share many properties with vertebrate NSCs, along with aspects of brain development, with all neurons and glia arising from a small set of progenitor cells [4]. *Drosophila* NSCs also have quiescent and reactivation states. In the embryo, NSCs form between 3.5h and 5.5h after fertilisation (a.f.) and generate progeny until the end of embryogenesis (17h a.f.) when they enter quiescence, a state where they remain until they are reactivated by dietary nutrients during early larval stages. An exception being four mushroom body NSCs per brain lobe, which continuously divide from embryonic stages to larval stages [4,10,11,12,13]. In the adult mammalian brain, the generation of new neurons and glia from NSCs is predominant in two major neurogenic niches, the ventricular-subventricular zone (V-SVZ) lining the lateral ventricles, and the subgranular zone (SGZ) of the dentate gyrus (DG) in the hippocampus [14,15,16]. NSCs in the V-SVZ, also known as radial glial-like, type B, cells, generate different interneurons of the mouse olfactory bulb and can give rise to glia, including oligodendrocytes and astrocytes, albeit in lower numbers in homeostasis. SGZ NSCs (also known as Type I cells) generate granule neurons, and at least a Nestin-expressing subpopulation has been reported to convert into astrocytes upon terminal differentiation. When cultured with growth factors, both V-SVZ and SGZ derived NSCs can generate the three neural lineages, highlighting their multipotency [5,7,17,18,19,20,21,22,23,24]. Another powerful model increasingly used to study NSCs is the zebrafish [25]. Compared to mammals, adult zebrafish have the incredible ability to heal injuries and repair damage within the CNS through the precise control of NSCs throughout the high abundance of stem cell niches [25,26]. In this review, we aim to bring together findings acquired using each model to highlight and discuss current perspectives on quiescence and reactivation of NSCs (Figure 1).

## 2. NSC Quiescent States, Activation and the Niche Influence

Quiescence is defined as a reversible exit from the cell cycle, with quiescent cells observed in the canonical G_0_ as well as in G_2_ states [4,10,35,36,37]. It is a functionally important cellular state to protect stem cells from proliferative exhaustion and maintain their longevity and self-renewal capacity for tissue homeostasis and repair. In addition, quiescence protects cells from environmental stresses such as hypoxia or nutrient starvation and allows them to avoid accumulation of DNA, protein and mitochondrial damage, which occur in dividing cells [5,27,36,38,39]. While quiescence is associated with a low metabolic rate, reduced translation, transcription and ribosome biogenesis, a growing body of evidence indicates it is an actively maintained state, involving signalling pathways to prevent senescence, suppress terminal differentiation, ensuring reversibility and re-entry into the proliferative cell cycle. Extrinsic signalling, conveyed by the cells’ niche, and intrinsic factors converge to regulate the transition between quiescence and proliferative active states [5,39,40].

Quiescence has been well defined in *Drosophila* NSCs, with temporal transcription factors and co-factors Pdm, Castor (Cas), Squeeze (Sqz) and Nab, involved in triggering quiescence, while the Hox proteins Antp and Adb-A spatially control quiescence entry [41]. *Drosophila* NSC (also known as neuroblast, NB) lineages are stereotyped, well described and named according to their position in the CNS [42]. Nab also functions in an alternate pathway in specifying temporal identity within the NSC 3-3T lineage (NB 3-3T, the NSC located in the third row and third column in each thoracic segment of the CNS ventral nerve cord), as NSCs retain their identity during quiescence [41]. Interestingly, it has been characterised that 75% of *Drosophila* qNSCs are arrested in G_2_ and only 25% in G_0_ [43]. In the CNS, the dorsal region primarily contains G_0_ NSCs, while the ventral contains primarily G_2_, which is facilitated by dorsal-ventral patterning cues [44]. These dorsal-ventral differences are induced by the dorsal patterning transcription factor Drop (Dr)/ Msx, which promotes p57/Dacapo (Dap) in the dorsal NSCs and thus inducing G_0_ quiescence [3,44]. Both Dr/Msx and p57/Dap are evolutionarily conserved, suggesting similar patterning-mechanisms may induce differing states of quiescence in other systems [44]. In ventral NSCs, NKX/Vnd is expressed and induces G_2_ quiescence. G_2_ qNSCs reactivate faster than G_0_ qNSCs and primarily express the evolutionarily conserved pseudokinase Tribbles (Trbl) [43]. Trbl inhibits insulin signalling by binding to Akt and preventing its phosphorylation, which along with the degradation of Cdc25^String^, maintains quiescence in G_2_ qNSCs [43]. Reversible G_2_ arrest through nutrient restriction has also been shown in *Xenopus laevis* tadpole NSCs, suggesting that G_2_ quiescence may be a cellular mechanism to control brain development in response to the availability of nutrients [45]. 

G_2_ quiescence is yet to be described in mammalian models but, both spatial and temporal patterning mechanisms appear to be present, as qNSCs in the V-SVZ line the ventricular wall while aNSCs reside in the periventricular region and area-specific regulation of quiescence has been reported [46,47]. The majority of NSCs reside in a G_0_ quiescent state in the adult mammalian brain reactivating to maintain homeostasis, responding to physiological cues and upon brain ischemia or injury [5,7,22]. Once reactivated, NSCs have the ability to self-renew and differentiate to generate new neural cells. Dysregulation of quiescence or reactivation of NSCs can result in depletion of their pool and decreased neurogenesis [5,7,22]. 

Comparable to G_0_ and G_2_ quiescent stages in *Drosophila*, quiescence in vertebrates is also not a single state but a continuum of varying degrees of depth [5,39,48]. Shallow qNSCs have a shorter reactivation time and are said to exist in a ‘GAlert state’, whereas deeper qNSCs require a longer period of exposure to physiological stimulus to reactivate [48]. In response to brain ischemia, a transition of deep qNSCs into a ‘primed’ quiescent state, in which NSCs are ready for rapid cell cycle re-entry, is promoted by interferon gamma signalling [27,48]. In mice, NSCs that have previously proliferated and return to a shallow quiescent state are said to be ‘resting’, compared to NSCs that have not yet proliferated which are termed ‘dormant’ [3,48]. NSCs in a shallow quiescence state are more likely to re-enter the cell cycle compared to NSCs which have not recently divided [48]. The proneural transcription factor Achaete-scute homolog 1 (Ascl1) is expressed in the V-SVZ and SGZ by active (aNSCs) and at a significantly lower level in qNSCs [5,48]. Despite low levels of Ascl1 expression in qNSCs, it is required to promote the transition of hippocampal NSCs from deep to shallow quiescence by modulating the activity of key signalling pathways [48]. Deletion of Ascl1 has been shown to move the population of qNSCs into a deeper quiescent state [48]. Through the transition from deep to shallow quiescence, Ascl1 induces the expression of Mycn via direct transcriptional activation, driving NSCs through shallow quiescence towards an activated state, via increased transcription and translation [48]. In Mycn KO mice, a greater percentage of NSCs are identified in a quiescent state, as qNSC cannot progress through shallow quiescence [48]. Zebrafish NSCs have also been shown to exhibit states of shallow and deep quiescence [49]. 

NSCs form an heterogenous population, coexisting in quiescent and proliferating states, showing region specific molecular properties in the mouse V-SVZ and giving rise to different progeny types [5,22]. This diversity is only beginning to be uncovered but increasing evidence highlights key signals that are received from, and exchanged with, the part of the neurogenic niche in which NSCs reside, heavily influencing their behaviour including their transition between quiescent and active states [5,22]. Different physiological stimuli impact NSCs and their niches. These cues include the presence of nutrients and feeding behaviour triggering hypothalamic-derived signals that control activation of a V-SVZ NSC subsets [50], and physical exercise such as running, which activates quiescent SGZ radial NSCs and has since long known to increase numbers of neurons in the hippocampal DG [51,52]. Pregnancy and early motherhood including the lactation stage have also been reported to activate V-SVZ NSC subpopulations, with specific olfactory neuron subtypes generated during the former [53,54]. Using live imaging in freely behaving mice, the day/ night cycle was shown to control V-SVZ NSC activation versus quiescence, a process dependent on darkness-induced circadian hormone melatonin signalling and favouring proliferation during day light [55], providing another example of physiological signals, which can be highly dynamic, influencing NSCs. In their niche compartments, adult NSCs respond to these and other stimuli as well as more local signals, interacting with the varied surrounding types of neural, vascular and immune cells, ECM and cerebrospinal fluid (CSF) in the case of the V-SVZ [5,6,7,22,56]. Neuronal activity, and in particular neurotransmitter and neuromodulators release by neuronal cells in the NSC niche, or located further away in the brain, can modulate NSC states, as exemplified by studies demonstrating GABAergic signalling limits proliferation [57,58]. Very recent findings show that a feedback mechanism also exists in which the proliferative state of V-SVZ NSCs is repressed by their rapidly dividing progeny of transient amplifying cells (TAPs), via Ephrin (Efn) signaling and Ca^2+^ dynamics [59]. This is consistent with studies in other tissue stem cells, such as by Walker and colleagues showing activation of mesenchymal stem cells of the mouse incisor to be coordinated by the respective TAPs [60]. The vasculature is another prominent niche component, with both NSCs and TAPs contacting directly endothelial cells [61]. Interestingly, while vasculature secreted factors have been shown to promote adult V-SVZ NSC lineage progression [62], diffuse endothelial and pericyte signals from the cortex were reported to be more potent at eliciting proliferation than those from the V-SVZ niche [63]. Moreover, direct cell–cell interactions between NSCs and endothelial cells, via exposure to EphrinB2 and Jagged1 ligands, have been demonstrated to restrict NSC proliferation and differentiation towards preventing stem cell pool depletion [64]. The studies suggest differential modulation of the vascular niche on NSC behaviour. 

During ageing, there is a decline in NSC function resulting in decreased production of neural cells, accompanied by reduced cognitive and olfactory discrimination [65,66]. This decline is thought to involve varied processes including an increase in the fraction of quiescent NSCs, which also enter deeper states of quiescence and have reduced ability to activate during homeostasis or upon injury [48,67]. Several age-related changes occur in NSCs affecting quiescence and activation [65]. Interestingly, the NSC functional decline is connected to an increase in glucose uptake, with transient glucose starvation or knockout of the GLUT4 glucose transporter Slc2a4, restoring old NSCs’ ability to activate [68]. Aging not only affects NSCs but also their niche components. Notably, the transcriptome of young and old NSCs was reported not to differ greatly, and old NSCs to function similarly to young ones in that they can generate similar progeny numbers [67]. These findings favour argument that age-related changes in NSC quiescence are at least in part due to niche alternations and signalling exchanged with it. In support, changes in inflammatory signals from the niche in aging brains have been linked to quiescence, with inhibition of acute inflammation in the old V-SVZ niche demonstrated to activate NSCs [67]. Adult NSC interactions with their niche in homeostasis and upon aging have been extensively reviewed [5,22,65,66]. In the next sections, we focus on reviewing molecular changes and signalling pathways underlying intrinsic and extrinsic factors controlling NSC quiescence and activation.

## 3. Alterations in mRNA Processing During Quiescence

In *Drosophila* qNSCs, messenger RNA and polyadenylated [poly(A)] RNA accumulate in high levels in nuclei in comparison to active aNSCs [69]. These alterations in nucleocytoplasmic portioning of poly(A) RNA and mRNA likely regulate the reduction of protein synthesis in qNSCs mentioned in the above section [40], but may also allow reactivation under the appropriate cues [69,70]. This trait of qNSCs is also present in adult mouse hippocampal NSCs, suggesting nucleocytoplasmic portioning might be an evolutionarily conserved hallmark of qNSCs [69,71]. Both *Drosophila* and mouse NSCs can be induced into quiescence when Nup98-96, nucleoporins involved in regulating the transport of macromolecules between the nucleus and cytoplasm, is reduced [69]. Other nucleoporins, including Nup155 and Tpr along with 204 out of 228 detected splicing factors are also downregulated [69]. One nucleocytoplasmic transport factor that may be downregulated in qNSCs of *Drosophila* is RanGEF/Rcc1/Bj1, which is thought to exclude the homeodomain transcription factor Prospero from NSC nuclei, thus its downregulation might enable the induction of quiescence by nuclear Prospero accumulation [72,73]. In mammals, nuclear transport is also altered in quiescent stem cells. The expression of Exportin-5 has also been shown to be down-regulated at protein level by autophagy, and at mRNA level by miR-34a, which may be facilitated by Exportin-1 [71]. Unlike Exportin-5, Exportin-1 is required for the processing of miRNAs that are induced through quiescence, allowing for pri-miR-34a to enter the nucleus and downregulate mRNA levels [71]. These Exportin-1 dependent changes have also been characterized in zebrafish NSCs, suggesting conserved parallels in mRNA biogenesis and localisation in quiescence across organisms [69,74]. In mammal, some miRNAs have been shown to play a role in regulating quiescence. For example, the overexpression of miR-221 and miR-222, which target p27 and p57 mRNA, was shown to induce S-phase bypassing quiescence when overexpressed [75].

Interestingly, in the study by Rossi and colleagues, 12 of the 228 detected splicing factors were detected upregulated in qNSCs, which may ensure the regulation of genes required for quiescence and priming NSCs for reactivation [69]. The differential expression of splicing factors suggests that there is a complex regulation of mature mRNA and immature poly(A)-RNA, facilitated through the alterations of many splicing and nucleocytoplasmic transport factors. This appears also to be the case in zebrafish, with one example being the nuclear localisation of miR-9 and Argonaut (Ago) proteins specific within qNSCs of the adult telencephalon, potentiating Notch signalling and targeting the FGF signalling pathway [76,77]. The localisation of nuclear miR-9 and Ago proteins is also conserved in the adult mouse brain [77]. Thus, alterations in nucleocytoplasmic portioning of poly(A) RNA and mRNA are likely to be a conserved mechanism inducing quiescence features and reducing protein synthesis in NSCs, whilst also priming them for reactivation under certain conditions.

## 4. Promotion and Maintenance of Quiescence Through Notch Signalling

The first *Notch* mutant was identified in *Drosophila* in 1913, and sequencing of the *Notch* gene in the mid 80s revealed it encoded a large transmembrane receptor-like protein. Since then, Notch signalling has been intensively studied, found conserved among species and with key roles in cell-cell communication involving cell fate specification, proliferation, differentiation and death [78,79,80,81]. There is, however, limited knowledge surrounding the involvement of Notch in *Drosophila* qNSCs. Only recently it has been shown that Notch signalling regulates central brain NSC quiescence, through the activation of the cell cycle exit gene *dap*, promoting G_0_ arrest [40]. Notch signalling plays an important role in maintaining adult mammalian NSC quiescence through cell-cell contact [5,7,82]. In response to dietary nutrients, low Notch levels are required to allow for the reactivation of qNSCs [40]. Notch ligands Delta (Dll1 and Dll4) and Jagged (Jag1) are expressed in the V-SVZ and SGZ by aNSCs and bind Notch receptors, activating the release of Notch intracellular domain (NICD) from the cell membrane to the cytoplasm [7,83]. NCID is then translocated to the nucleus where it forms a complex with DNA-binding CSL protein (Rbpj) and induces the expression of transcriptional repressor genes, such as hairy and enhancer of split 1 and 5 (*Hes1* and *Hes5*), which inhibit proneural genes such as Ascl1, thereby suppressing cell cycle re-entry and inhibiting neuronal differentiation [7,84]. Loss of Notch effector genes *Hes1* and *Hes5* or deletion of downstream mediator *Rbpj* results in activation and premature differentiation of NSCs and depletion of the NSC pool in the CNS of adult mice [5,82,83,84]. Overall, Notch is required for the maintenance of the NSC pool, by inducing NSCs in a quiescent state, whereas inactivation of Notch signalling leads to NSC proliferation and differentiation, depleting the NSC population [83]. Yet, expression of *Notch1*, *Notch2*, and *Notch3* varies between cellular states [7]. *Notch1* is highly expressed by proliferating NSCs and has been shown to promote proliferation of aNSCs without affecting qNSCs [5,46,82]. On the other hand, *Notch2* and *Notch3* are expressed to a higher extent in qNSCs, and deletion of *Notch2* and *Notch3* results in the activation of qNSCs and eventual depletion of the NSC pool [5,7,82]. 

In zebrafish, Notch plays a very similar role to the mammalian system, with a majority of functionally conserved orthologue ligands and receptors [25,28]. The pallial radial glial cells present in the telencephalon are similar both molecularly and cellularly to mouse adult NSCs. Here, Notch signalling is also a master regulator of quiescence and reactivation, with Notch3 controlling quiescence through *hey1* expression while Notch1/1b signalling is involved in injury response [28,85,86,87] Notch signalling controls the regenerative response to injury in the telencephalon [88,89]. Injury induces the proliferation of *olig2^+^* radial glia cells, likely through the inhibition of Notch, allowing for the regeneration of oligodendrocytes [88]. On the other hand, injury also activates Notch signalling in proliferating cells in the ventricular zone of the telencephalon, producing neuronal precursor cells that migrate to the injury site to differentiate into mature neurons, a process which may be partially conserved in mammals [89]. 

## 5. Bone Morphogenetic Proteins in Quiescence and Regeneration

Bone morphogenic proteins (BMPs) belong to the superfamily of ligands that take part in the Transforming growth factor-β (TGF-β) pathways, which have key roles during CNS development and adult function [90]. Interestingly, BMP signalling appears to diverge in promoting proliferation of NSCs in late larval stages of *Drosophila* versus contributing to NSC quiescence in the brains of adult mice [91,92]. If BMP is required for *Drosophila* NSC reactivation is yet to be determined. In adult mammalian NSCs, BMP induces NSC quiescence by transducing signals through Smad and non-Smad signalling pathways [93,94]. BMP4 in combination with FGF2 has been shown to induce a shallow quiescent state whereas BMP4 alone induces a deeper quiescent state [95]. In the mouse postnatal dorsal V-SVZ, single cell transcriptomics shows that subpallial NSCs remain primed for activation, whereas pallial NSCs enter a state of deep quiescence characterized by high BMP signalling, reduced transcriptional activity and Hopx expression. These findings highlight the heterogeneity of NSCs in the adult mammalian brain [96]. The two distinct levels of quiescence in those regions have been associated with the rapid closure of glutamatergic neurogenesis postnatally whereas gabaergic neurogenesis continues throughout life [96]. In the SGZ niche, BMPs produced by NSCs and granule neurons inhibit proliferation and maintain NSC quiescence by promoting the expression of inhibitory transcription factors such as Inhibitor of DNA Binding 1 (*Id1*)*, Id2* and *Id3* [7,93,94]. BMP type IA receptor (BMPR1A) is a key mediator of quiescence in adult NSCs as genetic deletion of *Bmpr1a* and *Smad4* results in short-term increased proliferation in the SGZ [94]. Also, the BMP antagonist Noggin inhibits BMP signalling by preventing the activation of BMPRs, allowing NSCs to re-activate, proliferate and differentiate [97]. However, long term deletion of *Bmpr1a* and *Smad4* and exposure to Noggin results in loss of NSCs regenerative capacity and depletion of newborn neurons [5,94].

In the zebrafish adult brain, BMP is primarily expressed in the neurons of the telencephalon, which communicates with radial glial cells that express the components of the canonical BMP signalling pathway, inducing *Id1* expression and promoting quiescence [25,98]. BMP signalling is induced in response to injury, indicating that BMP/Id1 has an essential role in balancing quiescence and reactivation in order to maintain the regenerative capacity of NSCs [98,99]. There may be links between the BMP and Notch signalling pathways, as both pathways have a positive effect on *her4.1* in expression within qNSCs, reinforcing the notion that the regulation of quiescence is a complex network of different signalling pathways acting together [98]. 

## 6. Hippo Signalling Maintains NSC Quiescence

Originally discovered in *Drosophila*, the Hippo pathway is one of the major signalling cascades that control organ growth and proliferation and is highly conserved throughout the animal kingdom [25,26,100]. The major kinase cascade within this pathway is composed of Hippo [MST1/2 in mammals], Salvador [SAV1], Wats [LATS1/2], Mats [MOB1] and Yorkie [YAP, TAZ] [2,8,100,101,102,103,104,105,106] and is regulated by numerous upstream inputs. The Hippo/Salvador kinase complex phosphorylates and activates the Wts-Mats kinase complex, leading to the phosphorylation of Yorkie, resulting in its cytoplasmic retention and degradation and thus maintenance of quiescence in both *Drosophila* and mammalian NSCs [8,32,102,103,104,105,106,107].

Several mechanisms regulate the Hippo pathway in larval qNSCs in *Drosophila*. For example, it can be activated by niche glial cells expressing Crumbs (Crb) and Echinoid (Ed) and the secretion of Anachromism (Ana) [107]. In response to nutrition, Crb and Ed are downregulated which inactivates the Hippo pathway, allowing Yki to relocate into the nucleus and initiate reactivation through activation of numerous genes in the NSC including the microRNA *bantam* [32,107]. Members of the conserved Striating-interacting phosphatase and kinases (STRIPAK) complex, which include Monopolar spindle-one-binder family member (Mob4), Connector of kinase to AP-1 (Cka) and the catalytic subunit of Protein phosphatase 2A (PP2A [Mts]), have been shown to orchestrate the Hippo and Insulin Receptor (InR/PI3K/AKT) signalling pathways [108,109,110]. PP2A/Mts has been shown to play different roles in quiescence and reactivation of early postembryonic NSCs [111]. During reactivation, Mob4 and Cka cooperate to associate PP2A/Mts with Hippo, dephosphorylating the Hippo kinase and inactivating the pathway allowing cells to reactivate [110,111,112]. PP2A/Mts and its regulatory subunit Wdb inactivate Akt, independent of Cka and Mob4, allowing Hippo signalling to remain active and thus maintain NSC quiescence [111]. Along with the regulation of the Hippo pathway, it has been reported that PP2A also dephosphorylates Cubitus interruptus (Ci) to increase Hedgehog (Hh) signalling, another pathway implicated in quiescence [113]. 

In the adult mammalian brain, the Hippo pathway is evolutionarily conserved and also active during quiescence [102,114,115]. Inhibition of the Hippo pathway via a variety of intrinsic and extrinsic factors allows NSCs to transition from a quiescent to active state [102,103]. Reduced Hippo pathway activity results in dephosphorylation and subsequent nuclear localisation of YAP/TAZ, which then bind with TEA domain transcription factors (TEADs) and promote the expression of genes that drive cell cycle entry and proliferation [8,104,105,106]. The Hippo pathway is tightly controlled, as overexpression of YAP does not result in NSC activation in the adult hippocampal SGZ, however overexpression of a mutant form of YAP (YAP-5SA; a protein insensitive to phosphorylation-dependent inhibition by LATs), promotes qNSCs in the hippocampus to re-enter the cell cycle in vivo and in vitro [102]. Analysis of single cell RNA-sequencing data reveals YAP highly enriched in aNSCs in the adult hippocampus, specifically in Sox2-positive cells of the SGZ [102]. YAP immunoreactivity co-localises with nuclear Sox2 expression in a subpopulation of these NSCs, suggesting NSC activation is accompanied by YAP nuclear translocation [102]. Furthermore, genetic deletion of YAP results in a significant reduction in the ratio of active versus qNSCs [102]. As mentioned in the above section, BMP signalling is one of the major factors involved in mammalian NSC quiescence. Specifically, BMP2 inhibits the proliferation of mouse NSCs through reduction of YAP nuclear translocation and YAP/TEAD interaction. The BMP2 effector Smad1/4 compete with YAP for the interaction with TEAD1 [105,106], linking Hippo and BMP pathways in maintaining qNSCs.

The Hippo pathway is also well characterised and conserved in zebrafish [116]. In this model Yap/Taz also promote regeneration [117]. *Yap1* and *wwtr1* [Taz] are both upregulated following spinal cord injury and localise to bridging glia [118]. To induce glial bridging, the ventral ependymoglia, CSF-facing cells with a radial morphology typical of the zebrafish CNS, undergo an epithelial-to-mesenchymal transition which has been linked to Yap1-Ctgfa signalling [117,118]. A better understanding of the role of the Hippo pathway in regeneration would be vital for the translation into the mammals and potentially provide therapeutical targets towards enhancing neural tissues regeneration. Mob4 has been shown to be highly expressed in the zebrafish central nervous system and implicated in neurodevelopment and cell proliferation [119,120]. PP2A has also been shown, along with p38 isoforms, to be expressed in quiescent Müller glia cells and in response to injury [121]. It was proposed that PP2A, along with Notch signalling, act downstream of the transforming growth factor TGFb3 to regulate Müller glia quiescence [121]; however little has been shown of its role in NSCs or radial glia cells.

## 7. Nutrient-Dependent Reactivation Through the InR/PI3K/Akt Cascade

The Insulin receptor InR/PI3K/Akt cascade is a highly conserved pathway that is regulated by insulin and involved in controlling growth during development and response to nutrient availability [122]. This pathway promotes reactivation of NSCs, with intake of nutrients, mainly amino acids, being its primary activator. In *Drosophila*, amino acids are sensed by the fat body, an organ serving roles of adipose tissue, liver and immune system, via detected by the amino acid transporter Slimfast (Slif, SLC7A in mammals), which activates the TOR pathway and generates a hormonal signal that stimulates PI3K/TOR signalling in the larval blood-brain barrier glia, inducing the release of Insulin like peptides (Ilps) [123]. Trbl, which is involved in G_2_ quiescence, is overridden by the nutrition-dependent secretion of dilp, enabling the reactivation of NSCs [43]. 

The secretion of Ilps by glial cells in *Drosophila,* or insulin-like growth factor 1 (IGF-1) from astrocytes, neurons and NSCs in the mammalian V-SVZ and SGZ recruits PI3K to the cell membrane and converts phosphoinositol(4,5)P2 (PIP2) to phosphoinositol(3,4,5)P3 (PIP3) [6,124]. PIP3 recruits several proteins including Akt, of which activation leads to reactivation of NSCs through numerous means, including the phosphorylation of FOXO (FOXO3 in mammals) excluding it from the nucleus and translocating it to the cytoplasm for degradation by ubiquitination [29,124,125,126]. FOXO3 mediates transcription of genes involved in G_1_/_2_ cell cycle arrest [8,126]. FOXO3 shares target genes with Ascl1 and suppresses Ascl1 target genes associated with cell cycle progression [5,127]. Therefore, FOXO3 inhibits NSC proliferation and maintains qNSC in the SGZ and V-SVZ by inhibiting Ascl1-mediated cell cycle entry [5,8,9,127]. Mice deficient in FOXO3 show an initial increase in NSC proliferation followed by premature exhaustion of the stem cell pool [9,29,126,128]. The PI3K/Akt phosphorylation cascade also leads to the activation of the mammalian target of rapamycin complex 1 (mTORC1) [29].

Milk-fat globule-epidermal growth factor (Mfge8) is secreted by SGZ NSCs, and astrocytes and promotes NSC quiescence via suppression of the PI3K-Akt-mTOR pathway. Mfge8 expression is enriched in qNSCs where it binds to Integrinβ1 and activates Phosphatase and tensin homolog (PTEN). PTEN negatively regulates PI3K activation, which inhibits Akt activation, therefore suppressing Akt-mediated activation of mTOR and NSC proliferation [5,7,92,128,129,130]. Loss of PTEN results in the activation of adult SGZ NSCs resulting in depletion of the NSC pool [5,9,92]. FOXO signalling has been well characterised between mammals and *Drosophila* [131,132]. In zebrafish, it is expressed in developing embryos and has also been implicated in promoting neurogenesis in the adult brain, with two isoforms of InR (*insra* and *insrb*) being highly conserved [133,134,135]. However, very little is known about TOR signalling in adult zebrafish NSCs.

Activation of the InR pathway in *Drosophila* NSCs also requires numerous other factors, including the physical association of the Heat shock protein 83 (Hsp83) and its co-chaperone Cdc37 in the presence of dietary amino acids [136]. Lack of amino acids leads to a large downregulation of *hsp83* mRNA, preventing the activation of the InR pathway in qNSCs. The Gap junction proteins Inx1 and Inx2 are also required in blood-brain barrier glia to ensure the translation of metabolic signals into synchronised calcium pulses in order to secrete Insulin [137]. 

One downstream target of the InR/PI3K/Akt pathway in *Drosophila* is Chromator (Chro), a spindle matrix protein critical in the reactivation of NSCs [138]. Chro is suggested to reactivate NSCs through activation of Grainyhead (Grh), and both inactivate the transcriptional repressor Pros [72,139]. Pros is an important differentiation factor involved in the balance between self-renewal and differentiation in NSCs, repressing cell-fate and cell-cycle genes, but has also been implicated in inducing NSC quiescence [72,139]. While high levels of Prospero trigger differentiation, low levels of nuclear Pros can promote quiescence by inhibiting the expression of Miranda, Asense and Cyclin E, arresting the cell cycle without triggering differentiation. Another gene, Fragile X protein (FMRP) has also been shown to regulate Cyclin E and phosphorylate Akt, which leads to decreased FOXO expression and premature exit from quiescence of NSCs [140,141]. 

## 8. SUMOylation and NSC Reactivation

SUMOylation, a type of post-translational modification that involves the conjugation of a small-ubiquitin-related modifier (SUMO), has been shown to have a role in the reactivation of NSCs [32]. *Drosophila* have a single SUMO gene *smt3*, which along with the SUMO conjugating enzyme 9 (Ubc9) promotes the SUMOylation of Hippo pathway Wts kinase at Lys766, inhibiting its activity and function and thus leading to reactivation [32]. During reactivation, Akt promotes the increase of Smt3 protein, suggesting that the InR/PI3K/Akt signalling cascade may promote the SUMOylation of Wts to inactivate the Hippo pathway [32]. In mammals there are five SUMO isoforms, but only SUMO1-3 are expressed in the brain [142]. Overexpression of Ubc9 in NSCs [12,32] elevates SUMO1-3 levels and increases the proportion of cells in the S-phase of the cell cycle, suggesting reactivation of qNSCs [142,143]. SUMO1 is also required for the activation of peroxisome proliferator-activated receptor γ (PPARγ), which regulates NSC proliferation [142]. On the other hand, loss of SUMO1 or Ubc9 increases Nanog, a transcription factor which promotes NSC self-renewal via the Hedgehog pathway, indicating SUMOylation has a negative effect on Nanog induced NSC self-renewal [142,144]. In addition, Protein inhibitor of activated STAT3 (Pias3)-induced SUMOylation of Sox2 further inhibits Nanog expression, impairing NSC self-renewal capacity [142,144]. Additional Sox transcription factors are modified by SUMOylation. Sox6 increases NSC self-renewal by upregulating B-cell lymphoma 2 (Bcl-2) and *Hes1* expression and enhancing Akt phosphorylation, but SUMOylation reduces its transcriptional activity [142,145]. SUMOylation has also been characterised in zebrafish NSCs. SUMOylating lysine 247 of Sox2 represses its expression, along with numerous targets involved in cell cycle control [146]. Ubc9 has also been implicated in zebrafish development; it is ubiquitously expressed during early development and restricted to proliferative zones at later stages, likely in stem cell niches [147]. However, SUMOylation has yet to be shown to be involved in NSC quiescence or reactivation in zebrafish. 

## 9. Lysosomal Activity and Autophagy in Quiescence and Reactivation

Lysosomal activity plays an important role in maintaining NSC quiescence in the V-SVZ of the adult mouse brain [148]. Enlarged lysosomes storing protein aggregates, higher expression of lysosomal genes, such as the lysosomal master regulator transcriptional factor EB (TFEB), and higher lysosomal activity have all been identified in qNSCs compared to aNSCs [129,148,149]. Nutrient starvation, used to induce quiescence in vitro, results in the dephosphorylation and activation of TFEB. Translocation of TFEB to the nucleus upregulates genes involved in lysosomal activity and degradation of activated EGF receptors (EGFR) [129,148]. Inhibition of lysosomal activity in vitro via deletion of TFEB, delays the reduction in EGF as well as Notch signalling, resulting in delayed NSC quiescence [129,148]. On the other hand, TFEB activation by Rapamycin inhibition of mTORC1, decreases aNSC proliferation [148]. Loss of TFEB in vivo prevents degradation of EGFR and Notch1, results in exit from quiescence and an increase in the number of aNSCs in the SGZ of young adult mice [148]. Conversely, elevation of lysosomal activity by ectopic expression of TFEB in aNSCs leads to quiescence induction [148]. qNSCs in the V-SVZ of aged mice express similar levels of lysosomal protein Lamp-1 to aNSCs [149]. TFEB activation in the V-SVZ of aged mice promotes reactivation of qNSCs, interestingly opposing the role of lysosomes on NSCs in the SGZ of young mice [129,148,149]. In zebrafish, increased expression of the lysosomal hydrolase activator Prosaposin (Psap) has been implicated in deep quiescence, likely enhancing lysosomal activity and autophagy [49].

Similarly, autophagy has been identified to contribute to shift the population of NSCs from a proliferative to quiescence state in the SGZ [150]. An increase in expression of autophagy genes (LC3-II and SQSTM/p62) as well as Aggrephagy receptor (Tax1bp1), highlights the upregulation of autophagy in qNSCs, potentially in response to the protein aggregates accumulated in enlarged lysosomes [129,148,150]. Furthermore, phosphorylation and activation of the AMP-activated protein kinase/unc-51 like autophagy-activating kinase 1 (AMPK/ULK1) pathway due to low availability sensed by AMPK enhances autophagy induced NSCs quiescence [150]. Loss of a critical gene for autophagosome formation, *Atg7*, in NSCs impedes the induction of quiescence during the development of the hippocampal DG NSC niche [150].

## 10. Metabolic Shifts Between NSC Quiescence and Reactivation

When NSCs transition from quiescent to active and vice versa, they switch metabolic pathways used to obtain energy. Lipid metabolism modulates Hedgehog signalling, another conserved signalling pathway, and is involved in reactivation in *Drosophila* [113,151,152], through the Lipid storage regulator droplet-2 (Lsd-2) and Fatty acid synthase 1 (Fasn1) [153]. In zebrafish, Hedgehog signalling has been linked to redox control, generating a positive feedback loop with H_2_O_2_ that contributes to regeneration [154]. These reports indicate that along with dietary amino acids leading to activation of the TOR and InR pathway, lipid intake and redox levels also play role in the reactivation of NSCs [12]. 

For mammalian hippocampal NSCs to reactivate from quiescence, lipid and fatty acid metabolism are downregulated and a transition to oxidative metabolism in the mitochondria must occur [29,37,155,156]. Increasing evidence, including via single-cell transcriptome analysis, points to different metabolic pathways being used to obtain energy in quiescent, ‘primed’ and aNSCs [27,29,155]. Fatty acid metabolism is enriched in qNSCs in the V-SVZ and downregulated upon NSC reactivation [155]. Fatty acid oxidation (FAO), the breakdown of fatty acids into acetyl-coenzyme A (CoA) in the mitochondria, is regulated by its rate limiting step, Carnitine palmitoyltransferase 1a (Cpt1a), which mediates the transport of fatty acids to the mitochondria [157]. Cpt1a is highly expressed in quiescent compared to aNSCs and blockage of FAO by a Cpt1 inhibitor in qNSCs resulted in cell death, highlighting the critical role of FAO in maintaining adult hippocampal NSC quiescence [157]. In addition, Thyroid hormone-inducible hepatic protein (THRSP/SPOT14) regulates lipid metabolism and is highly expressed in qNSCs [29,155,156,157]. SPOT14 downregulates the expression of Fatty acid synthase (Fasn), a key enzyme involved in de novo lipogenesis, by reducing the availability of malonyl-CoA, a fundamental building block in fatty acid synthesis [29,156,157]. Conversely, Fasn expression is elevated in proliferating NSCs as lipid synthesis is required for NSC division [29,157]. 

Glycolytic metabolism has also been highlighted as a characteristic of quiescent adult NSCs, with glycolysis genes such as *aldolase A*, *aldolase C* and *Lactate Dehydrogenase B (Ldhb)* shown to be enriched in qNSCs but significantly decreased upon reactivation [27,155]. FOXO3 plays a role in inducing quiescence-enriched metabolic genes, resulting in glycolysis [5,8,9]. FOXO3 is not only proposed to coordinate a program of genes in NSCs involved in quiescence and glucose metabolism but also related to aging and hypoxia, the latter including Hypoxia-inducible factor 1 (HIF1) targets [9]. Interestingly, conditional *Hif1a* gene deletion in adult mouse SVZ NSCs leads to depletion of the NSC pool, preceded by regression of SVZ vasculature, suggesting HIF1 as an essential NSC maintenance factor possibly via promoting vasculature niche integrity [158]. Downregulation of glycolytic metabolism is also observed in mammalian NSCs in a ‘primed’ quiescent state, prior to reactivation, suggesting glycolytic metabolism might be enriched in deep quiescence [27]. Interestingly, birth-associated downregulation of glycolysis and upregulation of glutamine metabolism linked to increased expression of *Glutamine synthetase (Glul)*, was recently reported to be required for embryonic mouse NSCs (radial glia, RG) induction into quiescence via suppression of mTORC1 signalling [159]. However, insufficient upregulation of Glul in preterm birth leads to transient hyperactivation of mTORC1 signalling in RG, and impairs transition from active to quiescent states, resulting in depletion of the NSC pool [159]. Depletion of the NSC pool can be prevented by suppressing the hyperactivation of RG in preterm birth mice with rapamycin treatment [159]. During the first postnatal week of DG development, NSCs are actively proliferating to expand the number of long-lived NSCs, prior to entering a quiescent state [160]. Suppressor of Fused (Sufu) regulates Sonic Hedgehog (Shh) signalling activity to control the production and maintenance of long-lived SGZ NSCs and the timely establishment of the qNSC pool [160]. Accordingly, deletion of Sufu or Shh receptor Smoothened (Smo) in NSCs or removal of Shh ligands during early DG development decrease Shh signalling, compromising postnatal expansion of long-lived NSCs, resulting in premature NSC quiescence and a diminished NSC pool in adult mice [160].

Shh signalling plays also critical role in the adult mammalian V-SVZ, balancing quiescent and activated NSCs by regulating quiescence. Deletion of the Shh receptor Patched (PTC) results in activation of the Shh signalling and an initial increase in proliferation of both qNSCs and aNSCs in the V-SVZ [161]. On the other hand, long-term activation of the Shh signalling pathway results in an accumulation of qNSCs and a progressive depletion of the aNSC pool [161,162]. In the SGZ, normal levels of Shh signalling by the mossy cells, a subgroup of neurons, acts to reactivate qNSCs. Under excitotoxic conditions, Shh signalling is upregulated, resulting in decreased NSC activation and an accumulation of qNSCs [162]. The authors suggest this is a protective mechanism to preserve the NSC pool, as qNSCs are more resistant to these toxic conditions due to their reduced rate of proliferation [162]. In contrast, activation of the Shh signalling pathway in vitro causes an increase in aNSC proliferation by shortening of both G1 and S-G2/M phases of the cell cycle. This leads to an increased number of cells generated and the potential of aNSCs to form colonies [161]. Treatment with an antagonist of the receptor Smo reduced the number and size of colonies formed by aNSCs in vitro [161]. 

## 11. Cellular Protrusions and Adhesion in Quiescence and Reactivation

One hallmark of *Drosophila* qNSCs is the extension of a primary, microtubule-enriched protrusion from the cell body towards the neuropile [124,132,138,163,164,165]. These protrusions have the ability to regenerate upon injury, requiring the Golgi apparatus to act as a major acentrosomal microtubule-organising centre [30,166]. Patronin, a microtubule minus-end binding protein, is an important factor within these protrusions, required for promoting acentrosomal microtubule growth, the regeneration of the qNSC primary protrusions upon injury and the reactivation of NSCs [30]. Both the promotion of acentrosomal growth and the reactivation of qNSCs are governed by Patronin, a Golgi-resident GTPase Arf1, and the microtubule polymerase Mini spindles (Msps) [30,166]. Patronin is distributed in the cytoplasm of qNSCs and can physically associate to the GDP-bound form of Arf1, which is regulated by its guanidine nucleotide exchange factor, Sec71 [30]. Arf1 and Sec71 are critical in microtubule assembly and orientation in qNSCs, and localise towards the protrusion initial segment (PIS), an indispensable region required for qNSC protrusions [166]. Msps also physically associates with Arf1, both in GTP- and GDP- bound forms, likely acting as an effector of Arf1 during reactivation [166]. Msps functions upstream of Kinesin-2 which transports and allows E-cadherin (E-cad) to localise to NSC-neuropil contact sites and leads to NSC reactivation [164]. 

In mammals qNSC protrusions are also highly enriched in actin. The polymerisation of fine filamentous actine (F-actin) structures promotes the movement of Myocardin-related transcription factor (Mrtf) into the nucleus, leading to the reactivation of NSCs [167]. F-actin polymerisation is mediated by the activation of the G protein-coupled receptor Smog on astrocyte-like glia in the NSC niche, signalling down a cascade of G protein α_q_ subunit, Rho1 and Diaphanous (Dia)/Formin. This results in the flow of F-actin patches, promoting polymerisation of F-actin and thus translocation of Mrtf into nucleus to promote the transcription of *actin* and other genes required for cell proliferation [167]. Cellular protrusions are also highly enriched in clusters of mitochondria, potentially linking the role of protrusions and metabolic shift in regulating quiescence and reactivation [168].

Primary cilia have been shown to serve as hubs for a multitude of signalling pathways, such as those regulated by Hedgehog (HH), receptor-tyrosine kinases, G-protein-coupled receptors, Wnt and TGFβ/BMP [169]. As discussed above, BMPs plays an important role in inducing quiescence in adult mammalian NSCs [93,94]. BMP4-induced qNSCs show an upregulation of genes associated with the development, organisation and movement of primary cilia, compared to BMP4 plus FGF2-induced qNSCs and aNSCs [95]. Cilia are key microtubule-enriched cellular protrusions acting as hubs transducing signals from the extracellular space [170]. Cilium extension is known to occur in G_1_/G_0_ phase of the cell cycle, and most long term qNSCs display a primary cilium in adult mice V-SVZ [95,171]. A key regulator of motile ciliogenesis, Forkhead Box J1 (Foxj1), is significantly upregulated in BMP4-only induced qNSCs compared to BMP4 plus FGF2-induced qNSCs, and its expression increases with the duration of BMP4 treatment [95]. Morphological differences in cilia have also been observed with primary cilia being longer in BMP4-induced qNSCs than BMP4 plus FGF2-induced qNSCs and aNSCs [95]. The longer cilium in BMP4-induced qNSCs are more developed and corresponds with a longer duration in a deeper quiescent state [95]. When cells re-enter the cell cycle, extended cilia are dismantled, therefore, the length of cilia is negatively correlated with cell cycle progression [68,171]. However, non-ciliated qNSCs have also been identified [171]. Ablation of primary cilia in a ventral domain of the V-SVZ and the SGZ disrupts the Shh signalling pathway, leading to decreased postnatal NSC proliferation and loss of the qNSC pool [172,173,174]. Moreover, knockout of *Rsph3a*, a mouse orthologue of human *RSPH3* for which mutations lead to primary dyskinesia—a rare condition resulting in structural or functional defects in motile cilia and sperm flagella [175] - causes increased activation of NSCs in aged mice [68]. In zebrafish, it has been shown that the conserved orthologues of Patronin, CAMSAP1-3, are present within active radial glial cells and involved with controlling microtubule growth [176,177]. The radial glial cells do have protrusions named cilium, which connect to the ventricular cavity and cerebral spinal fluid, which provides growth factors and a niche that is necessary for retaining their stemness [178]. Loss of these cilium through laser ablation leads to a delayed cell-cycle re-entry and disrupts asymmetric division, implicating their potential role in quiescence and reactivation [179]. Collectively, these findings suggest that different microenvironment-controlled effects can be mediated by cilia triggering signalling pathways on qNSCs and their activation. The dynamic microenvironment provided by extracellular matrix (ECM) molecules influences virtually all aspects of development and function of the CNS and its NSCs [180]. In agreement, a plethora of literature highlights the importance of stem cell adhesion and position within the niche as determinants of quiescent and active NSCs [47]. Transcriptome profiles of qNSCs have been shown to be enriched in gene ontology categories associated with cell adhesion [181,182,183]. Several genes exhibit preferential transcription in qNSC compared to aNSCs in the V-SVZ and SGZ, such as Neural cell adhesion molecules 1 and 2 (NCAM1, NCAM2), Vascular cell adhesion molecule (VCAM-1) and N-cadherin [3,6,155,182,183,184]. Loss of N-cadherin in the V-SVZ results in increased activation and proliferation of qNSCs, and depletion of the NSC pool [185]. Membrane-type metalloproteinase MT5-MPP regulates N-cadherin-mediated adhesion, promoting qNSCs activation [185]. Loss of VCAM1 induces qNSC reactivation and eventual NSC pool depletion [47,182,184,186]. VCAM1 is down regulated upon NSC exit from quiescence [182,184] and its deletion disrupts niche structure as well as leading to impaired hippocampal learning and memory [182,184,186]. WNT signalling is critical for the maintenance and differentiation of different types of stem cells [187]. Non-canonical Wnt signalling has been proposed as a key pathway in maintaining NSC quiescence in the SVZ, by activation of Rho-GTPase Cdc42, which maintains NSC adhesion to the apical niche [47]. Other studies have implicated canonical Wnt/β-catenin signalling. For example, NSC lineage tracing using an inducible Cre Axin2 mouse strain [188] showed that a subset of adult SVZ NSCs, derived from embryonically labeled *Axin2^CreERT2+^* precursors, are regionally and functionally restricted and retain Wnt/β-catenin responsiveness [189]. However, recent findings show that while both qNSCs and aNSCs express and can respond to components of the Wnt/β-catenin signalling pathway, it is dispensable for the maintenance of adult NSCs in a proliferative or quiescent state, and also for the transition of NSCs between quiescent and active states [190]. The authors comment that the apparent disparate results may be likely due to differing techniques used to modulate Wnt/β-catenin signalling in previous literature, as well as the use of in vivo and in vitro models, which may lead to different responses of NSCs. To understand if and when Wnt/β-catenin signalling has a role in NSCs, novel in vivo tools are needed allowing manipulation of Wnt gradients in vivo. Interestingly, as part of the canonical Wnt pathway, Wnt ligands bind to the class F (Frizzled) family of G-protein-coupled receptors, which localise in the primary cilia, however the functional relevance of this localisation is still unclear [169].

## 12. Mechanisms Behind qNSC Reactivation upon Injury

During injury, it is important to recruit and reactivate qNSCs to mount a regenerative response. In the *Drosophila* larval CNS, a coordinated neuro-glial response occurs upon injury [191]. Swim, a lipocalin-like transporter, is secreted from the glia in a HIF1- *α*-dependent manner, either through direct activation or indirectly through other hypoxia-responsive factors, which promotes the distribution of Wingless (Wg)/Wnt in the local tissue environment [191]. Increased levels of Wg/Wnt signalling promotes qNSCs reactivation, thus mounting a regenerative response. Interestingly, the mouse Swim orthologue, Lcn7, has been shown to be induced by glial cells in the hippocampus upon brain injury, suggesting a conserved function [191]. 

NSCs and astrocytes of the adult mammalian neurogenic niches have been reported to secrete Wnts, such as Wnt3a and Wnt5a [5,17]. The response of NSCs to Wnt/β-catenin signalling is a current topic of investigation. Wnt activation seems to be dose-dependent and state-specific [190]. As mentioned above, non-canonical Wnt signalling has been proposed to maintain NSC quiescence in the SVZ by activation of Rho-GTPase Cdc42, which regulates Notch signalling activity [47]. However, in response to demyelination injury, non-canonical Wnt signalling is downregulated, and activation of canonical Wnt/β-catenin signalling is reported to occur to re-activate qNSCs, which divide symmetrical to increase the stem cell pool for tissue repair [7,29,47,190]. Wnt3a induces the expression of a canonical Wnt/β-catenin target, Axin2, whereas Wnt5a antagonises the Wnta3a-induced Axin2 levels, suggesting non-canonical Wnt signalling overrides canonical Wnt/β-catenin signalling in homeostatic conditions [47,189,190]. Therefore, a transient shift from non-canonical to canonical Wnt signalling may be required for the activation of qNSCs for brain repair [5,47].

T cell factors (TCFs) are main transcriptional mediators of Wnt signalling, which have also Wnt independent activities [192]. In contrast to TCF proteins in *Drosophila* and *C. elegans* that are encoded by a single *Tcf* gene, vertebrates have different TCFs that likely increase context-specific responses to Wnt signaling. Studies of vertebrate embryonic development reveal that TCF3 acts mainly as transcriptional repressor whereas TCF1, lymphoid enhancer factor 1 (LEF1) and TCF4 activate transcription [192]. Interestingly, a TCF switch has been reported in which TCF3 repressor is replaced by TCF1 activator to regulate the homeobox *Vent2* gene expression in developing *Xenopus* embryos. It remains to be demonstrated if a similar model acts in mammals [192,193]. How TCFs act in NSC quiescence and activation is also not fully understood. Using a TCF/LEF-reporter expressing an H2B-GFP fusion under the control of TCF/LEF binding sites [194], Kalamakis and colleagues argue that canonical WNT activity is predominant in presumably qNSCs (Ki67-negative) of the dorsal domain, as compared to the latero-ventral domain, of the V-SVZ [67]. They hypothesize that a switch to non-canonical WNT is required for NSC activation throughout life, in contrast to studies proposing non-canonical WNT signalling to maintain NSC quiescence (e.g., [47]). As mentioned previously, the authors show that qNSCs increase in the old brain and are more resistant to leave quiescence even upon injury. Interestingly, block of sFRP5, a known WNT antagonist found uniquely expressed in old qNSCs, is shown to increase active NSCs in the old brain in both homeostasis and following temozolamide-induced injury, which may contribute to explain age-related overall decline in active NSC function and regenerative capacity [67]. LEF1 and its partner β-catenin have also been found accumulating in quiescent hippocampal NSC cultures, concomitant with enhanced TCF/LEF1-driven transcription and preponderance of a full length LEF1 isoform contributing to its stability [195]. Interestingly, LEF1 interacts with the transcription factor NFIX, however, if these interact at specific enhancers in quiescent NSCs to regulate the cells’ state requires further study [195].

Unlike *Drosophila* and mammalian NSCs, zebrafish are highly adept at healing injuries and repairing damage to their CNS, but the Wnt/ β-catenin signalling seems not required to reactivate qNSCs for regeneration [196]. As described above, Notch signalling has been shown to regulate regenerative response in the telencephalon [88,89], however BMP/Id1, Hedgehog and YAP/TAZ signalling have also been shown to be regulated following injury [31,117,118]. Leukotriene-C4 (LTC4) is reported to be sufficient to activate the regenerative response by inducing gata3 which promotes proliferation, along with the chemokine receptor Cxcr5, which shows increased expression after injury [87]. Another factor induced by injury in zebrafish is Islet Antigen-2 (Ia-2), which is highly conserved and involved in dense core vesicles to release Insulin, thus activating the InR signalling cascade [197]. Thus, the regenerative capabilities of zebrafish are regulated by numerous factors and pathways, many of which are present in mammals.

One potential reason the mammalian system lacks the regenerative capacity of zebrafish may be the presence of the tumour suppressor gene *ADP-ribosylation factor (ARF)*. ARF is important in suppressing cancer, but zebrafish, along with other organisms with the ability to regenerate, lack this gene [198]. Expression of human ARF in zebrafish was shown to localise into the nucleus along with Mdm2, stabilising Tp53 and inducing cell cycle arrest/apoptosis, suppressing fin regeneration [198]. This suggests that organisms with the ability to regenerate use widely conserved signalling pathways, but that mammals may express factors suppressing these signals. To potentially unlock the regenerative capacity in mammals without promoting cancer would be a major step towards tackling the rising burden of CNS injury and neurodegenerative diseases.

## 13. Conclusions

The research into the regulation of quiescence and reactivation of NSCs has been extensive across different model organisms. Together with supporting technology advance, it allowed better understanding of the complex network of signalling pathways and molecular mechanisms that regulate the critical balance between quiescence and reactivation. In this review, we have brought together findings using *Drosophila*, mammals and zebrafish models (Figure 2, Table 1). While much has been uncovered, there is still a long way to go including exploring the role of epigenetics in the control of NSC quiescence, which only recently has been entered into [199]. 

We only started to unravel the complex network of signalling pathways, epigenetics, genetic, and metabolic alterations; all contributing to the delicate balance that sustains the precious amount of NSCs required for development and repair. A detailed knowledge of how these cells maintain their states and depths of quiescence, how reactivation under certain conditions occurs and how to unlock the hidden regenerative potential that many regenerative organisms utilise are all critical milestones for the development of future regenerative therapies and combat the ever-rising burden of neurodegeneration.

## Figures and Tables

**Figure 1 biomolecules-15-00672-f001:**
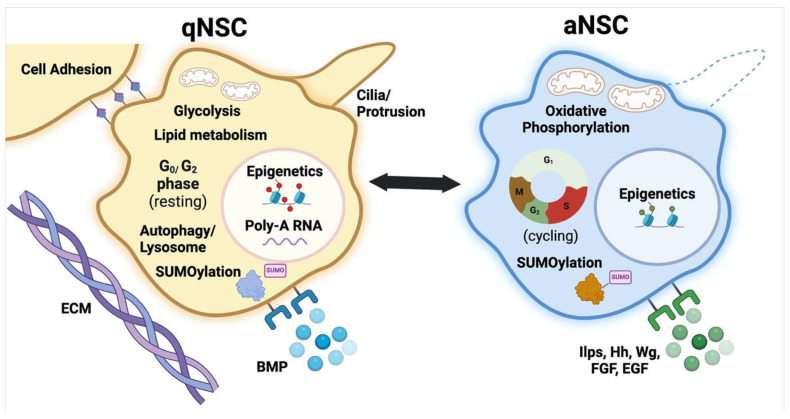
Several main features of quiescent and active NSCs. Alterations in metabolism, cell adhesion, Extracellular matrix (ECM), cilia/ protrusion, autophagy/ lysosomes, epigenetics regulation, SUMOylation, poly-A RNA nuclear accumulation and several ligand-receptors are depicted [6,12,17,27,28,29,30,31,32]. qNSC: quiescent NSC; aNSC: active NSC. Adapted from [33]. Created in BioRender [34].

**Figure 2 biomolecules-15-00672-f002:**
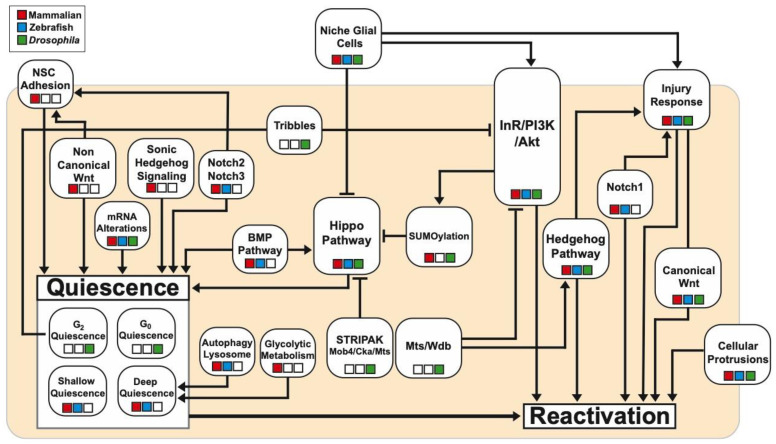
Key signalling pathways and processes involved in quiescence and reactivation of NSCs discussed in this review, and their interactions identified in *Drosophila,* mammalian or zebrafish models. Colour-coded boxes indicate of which model the data has been derived from. Arrow: activation; Bar: repression.

**Table 1 biomolecules-15-00672-t001:** Summary of findings and publication references discussed in this review using *Drosophila*, mammalian and zebrafish NSCs as model systems.

	*Drosophila*	Mammals	Zebrafish
Quiescent states, activation and niche influence	G0 and G2 Quiescence: [3,41,43,44]	Shallow and Deep Quiescence: [3,5,7,25,27,35,36,37,39,40] [46,47,48,160]; Niche Influence: [22,51,52,53,54,55,56,57,58,59,60,61,62,63,64,158]; Activation and Aging: [65,66,67,68]	Shallow and Deep Quiescence: [49]
mRNA Alterations	Nucleocytoplasmic portioning of poly(A) RNA and mRNA regulates qNSC and prime for reactivation: [69,72,73]	Nucleocytoplasmic portioning of poly(A) RNA and mRNA regulates qNSC and prime for reactivation: [69,70,71,75]	Nucleocytoplasmic portioning of poly(A) RNA and mRNA regulates qNSC and prime for reactivation: [74,76,77]
Notch Signalling	Regulates quiescence: [40]	Varied Notch1, Notch2 and Notch3 in quiescence and reactivation: [5,7,46,82,83,84]	Varied Notch1, Notch2 and Notch3 in quiescence and reactivation: [25,28,85,86,87,88,89]
BMP/Id1	Not known in NSC quiescence/ activation regulation.	Induces quiescence including through expression of Id1-3: [92,93,94,95,96,97]	Induces quiescence through expression of Id1; linked to Notch: [25,98,99]
Hippo Pathway	Maintains quiescence through Hippo/Salvador/Wats/Mats/Yorkie cascade: [2,32,100,107]	Maintains quiescence through MST1-2/SAV1/LATS1-2/MOB1/YAP/TAZ cascade: [2,8,100,102,103,104,105,106,114,115]	Maintains quiescence through Mst2/Sav1/Lats1-2/MOB1/Yap/Taz cascade: [25,26,116,117,118]
STRIPAK	Mob4 and Cka associate PP2A/Mts to inactivate Hippo Pathway PP2A/Mts and Wdb inactivate InR/PI3K/Akt pathway: [108,109,110,111,112,113]	Not implicated in quiescence or reactivation of NSCs.	Present within quiescent Müller glia cells and in response to injury, but limited knowledge in NSCs: [119,120,121]
Insulin Pathway	InR/PI3K/Akt/TOR pathway leads to reactivation of NSCs, including through inhibition of FOXO: [6,43,122,123,124,136,137,138,140,141]	InR/PI3K/Akt/mTOR pathway leads to reactivation of NSCs, including through inhibition of FOXO3: [5,7,9,29,92,125,126,127,128,129,130,131,132]	Limited knowledge. insra and insrb highly conserved: [133,134,135]
SUMOylation	smt3 and Ubc9 SUMOylate Wts leading to reactivation: [32]	Implicated in self-renewal and differentiation of NSCs: [142,143,144,145]	Ubc9 is present in early development and proliferative zones at later stages. Little known in NSCs: [146,147]
Lysosomal and Autophagy	Limited knowledge.	Lysosomal activity and autophagy are involved in maintaining NSC quiescence: [129,148,149,150]	Prosaposin (Psap) has been implicated in deep quiescence in zebrafish: [49]
Metabolism	Early lipid intake correlates to reactivation, modulated by Hedgehog: [113,151,152,153]	Fatty acid metabolism enriched in qNSCs; switch to oxidative metabolism required for reactivation; birth is associated with metabolic changes: [5,8,9,27,37,160] [29,155,156,157,159]	Limited knowledge; positive feedback loop of H2O2 contributes to regeneration through Hedgehog: [154]
Cellular Protusions and Adhesions	Cellular protrusions are hallmark of qNSCs; promotes reactivation through Golgi apparatus and Actin: [30,124,132,138,163,164,165,166,167,168,169]	Primary cilia are a hallmark of deep qNSCs; adhesions are highly important in determining quiescence or reactivation: [3,9,46,47,68,95,170,171,172,173,174,175,181,183,184,185,186,190]	Limited knowledge. Patronin is conserved in active radial glial cells; cilia are present and involved in retaining NSC stemness: [176,177,178,179]
Injury/Regeneration	Neuro-glial response coordinated by injury. Promotes Wg/Wnt distribution: [30,191]	NSCs responsive to canonical WNT signalling; Non-canonical Wnt signaling implicated in quiescence; TCFs; Canonical Wnt signaling required for brain repair: [5,7,29,47,67,189,190,194,195]	Highly adept at regeneration. Notch, BMP, Hedgehog, YAP/TAZ and InR pathways implicated in repair; ARF likely suppresses regeneration within mammalian system: [31,87,88,89,117,118,196,197,198]

## Data Availability

No new data were created or analyzed in this study.

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
