# Peer review of "The Critical Balance Between Quiescence and Reactivation of Neural Stem Cells"

_biomolecules, 2025, doi:10.3390/biom15050672_

Round 1
Reviewer 1 Report
Comments and Suggestions for Authors
The manuscript by Elkin and colleagues provides a comprehensive review about quiescence and reactivation in neural stem cells. It summarises recent findings from Drosophila, zebrafish and mammalian model systems while highlighting conserved genetic and molecular mechanisms regulating neural stem cell state transitions.
The review is well structured and well written. It is interesting for the naïve as well as the more knowledgeable readers in the field. The two figures and the table support the text and give a very nice overview about progress in the field. In especially figure 2 illustrates the findings in an innovative way.
Minor remarks to revise:
Page 3 line 84: the naïve reader might not know what NSC3-3T lineage means. The authors could explain that lineages are stereotype and very well described in Drosophila. They can be identified and are labelled according a coordinate system.
Page 5 line 228: Marcy et al., 2023, DOI: 10.1126/sciadv.abq7553 is an interesting study that could be added in this context.
Page 9 line 422: It seems that [96] is not the correct reference here.
Page 9: line 442: give primary references.
Reviewer 2 Report
Comments and Suggestions for Authors
The review by Elkin and coworkers described a detail analysis of extrinsic and intrinsic signals involved in the delicate balance between quiescence and reactivation of Neural Stem Cells (NSC). It provides an overview of the complex mechanism and pathways at play in three model systems drosophila fruit fly, mammalian adult brain, and zebrafish, within the longer term prospective of NSC based therapies, directed at tissue regeneration, repair and restoring functions.
In a first part an overview of the different states of quiescence, which exhibit varying degree of depth, and the genetic programs that maintain the quiescent state, prevent senescence, suppress terminal differentiation and allow the reversible reentry into the proliferative cell cycle are summarized.
The authors detail alterations of mRNA processing, the role of Notch and BMP signaling in balancing NSC quiescence and proliferation, providing details on impacts of Hippo signaling and its interaction with the Sonic HedgeHog (SHH) and BMP in maintenance of quiescence.
Reactivation of NSC through Nutrient influx mediated by the InR/PI3K/Akt cascade, or SUMOylation of signaling cascades intermediates, including Hippo and Pi3K/Akt, or the transcription factor Sox2 are described.
Finally, the roles of Lysosomal Activity and autophagy, and of metabolic shifts, including the impact of SHH pathway, and of cellular protrusions, including cilium, are detailed. The authors end their review by describing the mechanisms behind quiescent NSC reactivation upon injury and provide a complete scheme, and a table, summarizing the findings and related references for each of these chapters.
Overall, this review is well written, very extensive and extremely interesting. I will have a small remark regarding the WNT canonical pathway where the author mainly rely on the paper Austin et al 2021 (ref 158). They should also mention the role of T Cell factors (TCF) in NSC. TCF are indeed described as the main transcriptional mediators of WNT signaling but would also have WNT independent activities and maintain NSC throughout development (Development 138, 4341-4350 (2011) doi:10.1242/dev.066209 Bowman, A. N., et al . (2013). PNAS 110, 7324–7329.)

Reviewer 3 Report
Comments and Suggestions for Authors
The authors provide extensive coverage of molecular processes regulating neural stem cell activation, across three major in vivo animal models. The "evolutionary" context being provided is particularly useful. While assessing the completeness of the molecular/mechanistic descriptions provided is beyond what my core expertise (quantitative lineage dynamics) would allow, the coverage is very extensive and the supporting references can be assumed to provide ample opportunity for further reading to interested readers.
What seems to be mostly missing is a focused discussion of niche/environmental control of NSC activity. Obviously the numerous mentions of the role of secreted factors in controlling NSC activity do cover the topic at the molecular level. However, work e.g. from the Saghatelyan (circadian control of NSC activation, or more recent work on NSC interactions with their progeny and other cellular components of the niche) or Doetsch (various forms of control of NSC activity) labs are not (or scarsely) discussed. While some of these contributions may not have dissected molecular mechanisms or directly focused on q/aNSC transitions, discussing this and related work may provide a slightly "higher level" discussion of means and potential impacts of controlling NSC activity. The review is however already extensive and very focused on molecular details, so the above should really be intended as a suggestion rather than a requirement.
The authors mention NSC exhaustion a few times, mostly to discuss the impact of altered activity of signaling processes controlling NSC activity, but seem less focused on the physiological relevance of shifts in NSC (in)activation processes, as e.g. as impacting neurogenesis during with age (though Kalamakis et al. are referred to regarding changing responses to injury), which may be worth referring to.
Minor note: the authors refer twice to "mammalians", while I think "mammals" would work better in these cases (adjective being used as noun vs actual noun). There is one mention of "epidermal" radial cells (line 289), where I assume authors refer to ependymoglia, CSF-facing cells with a radial morphology typical of the Zebrafish CSN.
